

# 1 A model of the within-population variability of budburst in forest 2 trees

Jianhong Lin[1, *], Daniel Berveiller[1], Christophe François[1], Heikki Hänninen[2, 3], Alexandre Morfin[1],
Gaëlle Vincent[1], Rui Zhang[2, 3], Cyrille Rathgeber[4], Nicolas Delpierre[1, 5, *]
[1] Université Paris-Saclay, CNRS, AgroParisTech, Ecologie Systématique et Evolution, 91190, Gif-sur-Yvette, France.
[2] State Key Laboratory of Subtropical Silviculture, Zhejiang A&F University, Hangzhou, China
[3] SFGA Research Center for Torreya grandis, Zhejiang A&F University, Hangzhou, China
[4] INRAE, SILVA, Université de Lorraine, AgroParisTech, Nancy, France
[5] Institut Universitaire de France (IUF)
[*] *Correspondence to*: jianhong.lin@universite-paris-saclay.fr, nicolas.delpierre@universite-paris-saclay.fr
**Abstract.** Spring phenology is a key indicator of temperate and boreal ecosystems' response to climate change. To
date, most phenological studies have analyzed the mean date of budburst in tree populations while overlooking the
large variability of budburst among individual trees. The consequences of neglecting the within-population variability
(WPV) of budburst when projecting the dynamics of tree communities are unknown. Here, we develop the first model
designed to simulate the WPV of budburst in tree populations. We calibrated and evaluated the model on 48,442
budburst observations collected between 2000 and 2022 in three major temperate deciduous trees, namely, hornbeam
(*Carpinus betulus*), oak (*Quercus petraea*) and chestnut (*Castanea sativa*). The WPV model received support for all
three species, with a root mean square error of $5.6 \pm 0.3$ days. Retrospective simulations over 1961-2022 indicated
earlier budburst as a consequence of ongoing climate warming. However, simulations revealed no significant change
for the duration of budburst (DurBB, i.e., the time interval from BP20 to BP80, which respectively represent the date
when 20% and 80% of trees in a population have reached budburst), due to a lack of significant temperature increase
during DurBB in the past. This work can serve as a basis for the development of models targeting intra-population
variability of other functional traits, which is of increasing interest in the context of climate change.
Keywords: budburst variability; model; temperate trees; climate warming; budburst duration; population.

## 26 1. Introduction

Phenology, as the study of recurrent biological events such as budburst in spring, has attracted increasing attention due
to climate warming (Piao et al., 2019). The timing of leaf phenology in spring is a major indicator of climate warming
(Parmesan and Yohe, 2003) and is mainly modulated by temperature (Menzel et al., 2006; Zhang et al., 2022; Zhang
et al., 2021; Chen et al., 2018; Vitasse et al., 2009a) and photoperiod (Delpierre et al., 2016; Fu et al., 2019; Vitasse
and Basler, 2013; Meng et al., 2021). In the northern hemisphere, it is well established that spring phenological events
have been advanced by climate warming (Walther et al., 2002; Menzel et al., 2006), although this advancement is
currently slowing down (Fu et al., 2015; Chen et al., 2019). To date, massive efforts have been made to study the
spatiotemporal variability of leaf phenology among tree populations and across years (Delpierre et al., 2016; Fu et al.,
2015; Meng et al., 2021; Chen et al., 2018). However, the variability of leaf phenology within populations has received

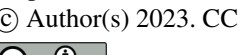



little attention to date (Scotti et al., 2016; Delpierre et al., 2017), which is in line with the general focus of ecological
studies on average traits (Violle et al., 2012). This is intriguing, since the within-population (i.e., tree-to-tree) variability
of phenological events is vast and can even be equivalent to that observed among populations (Delpierre et al., 2017;
Vitasse et al., 2009a; Rathgeber et al., 2011). It typically takes 1 to 4 weeks from the first to the last tree to burst buds
in a population (Denechere et al., 2021), with an average of 19 days (Delpierre et al., 2017). Furthermore, the duration
from the first to last tree to burst buds in a given population varies annually (Denechere et al., 2021).
The large within-population variability (WPV) of budburst observed in natural tree populations is considered to result
from their exposure to a large range of fluctuating environmental (e.g., frost) and biotic (e.g., herbivores and pathogens)
selection pressures, which alternatively favor trees that burst buds early or late (Alberto et al., 2011). From an
evolutionary point of view, this phenotypic diversity has an adaptive value at the population scale, because the
environment is likely to change across the lifetime of trees (Petit and Hampe, 2006; Morente-Lopez et al., 2022;
Blanquart et al., 2013). For instance, if a local climate becomes suitable in early spring under climate warming, trees
that burst buds early will benefit from an extended growing season, thus maximizing their carbon assimilation and
possibly their biomass production (Zohner et al., 2020; Delpierre et al., 2009; Richardson et al., 2010), which will
allow them to gradually occupy a dominant position in the population. Moreover, early budburst enables trees to escape
pathogens (e.g., for oak, see Dantec et al., 2015). On the contrary, if freezing events occur frequently in early spring
with the advance of budburst, late trees can grow better by avoiding freezing injury (Delpierre et al., 2017; Zohner et
al., 2020; Puchalka et al., 2016). Moreover, the WPV also affects interactions with competing plants and herbivores
(Hart et al., 2016; Renner and Zohner, 2018).
The WPV of budburst is probably underpinned by genetic diversity, as evidenced by the variability of phenological
traits among individual trees that experience similar environmental conditions (Bontemps et al., 2016; Delpierre et al.,
2017). This genetic determinism is further reflected in the year-to-year repeatability of the phenological ranking of
individuals within tree populations (Delpierre et al., 2017). In addition to this genetic determinism, the WPV is also
likely influenced by micro-environmental variations such as the unbalanced distribution of soil-water content within
populations, edaphic conditions, or microtopography (Delpierre et al., 2017; Denechere et al., 2021; Scotti et al., 2016).
To the best of our knowledge, the question of whether and to what extent would the WPV of budburst be modified in
the current context of climate change has not been addressed so far. Quantifying WPV as the duration (in days) from
the first to the last tree to burst buds in one population, we identify three alternative hypotheses for the modification
of WPV with climate change: (i) the duration of budburst remains unchanged because all trees in the population
advance to the same extent; (ii) the duration of budburst decreases because of the increasing warming rate during the
budburst period (Malyshev et al., 2022); (iii) the duration of budburst increases because of insufficient chilling
accumulation, as hypothesized previously from experimental studies (Zohner et al., 2018; Zhang et al., 2021).
Phenological research has made extensive use of modeling to study the response of the spatiotemporal variability of
budburst to climate warming (Zhang et al., 2022; Meng et al., 2021; Delpierre et al., 2009; Chuine and Regniere, 2017).
The models postulate that temperature and photoperiod are the main environmental cues that trigger budburst in boreal
and temperate (Delpierre et al., 2009; Kramer, 1994; Hänninen and Kramer, 2007), subtropical (Zhang et al., 2022; Du



et al., 2019), and tropical trees (Chen et al., 2017). In these models, temperature plays a dual role. Endodormancy is
released by chilling in late autumn or winter, with high temperatures allowing for ontogenetic growth during the
ecodormancy stage (Hänninen, 2016; Jewaria et al., 2021). Meanwhile, there is an interaction between these two stages
in the models, namely, ontogenetic growth is influenced by dormancy release (Hänninen, 2016; Hänninen and Kramer,
2007; Vegis, 1964). Lundell et al. (2020) further proved that this interaction can be affected by prevailing temperatures.
One important point is that these models do not pay attention to the WPV of phenological traits. They have been
parameterized and applied to predict the mean or median date of budburst in a given tree population (Lundell et al.,
2020; Kramer, 1994; Zhang et al., 2022). In other words, these models simulate the timing of budburst as a discrete
event in the population without considering the WPV of leaf phenology. To the best of our knowledge, only two studies
to date, notably (Rousi and Heinonen, 2007) in Birch (*Betula pendula*) and (Langvall et al., 2001) in Norway spruce
(*Picea abies* (L.) Karst.), have attempted to establish a link between WPV and environmental conditions through the
temperature sum required for the opening of buds at the scale of individual trees. At the scale of tree populations, a
distribution of temperature sums to budburst was also used in the so-called physio-demo-genetic (PDG) models
(Kramer et al., 2008; Oddou-Muratorio and Davi, 2014) to simulate the adaptive potential of tree populations. However,
a systematic model for the WPV of budburst is still lacking.
Here we developed a model that simulates the WPV of budburst in temperate deciduous trees. We calibrated and
validated the model over an extensive budburst dataset acquired from five tree populations at the individual tree scale
over 23 years (representing 48,442 observations). Specially, we aim to 1) develop the WPV model and validate its
ability for predicting the progress of budburst in tree populations, 2) use the model to in a retrospective simulation
exercise testing whether the duration of budburst period in the population changed with climate warming in the recent
decades.
## 2.   Materials and Methods
### 2.1 Study sites
We used budburst data collected from two forests located near Paris (France): Barbeau (48.476° N, 2.780° E, 95 m asl)
and Orsay (48.705° N, 2.167° E, 105 m asl). At these sites, the progress of budburst was observed at the individual
scale in populations of three major temperate deciduous tree species, namely, hornbeam (*Carpinus betulus* L.), oak
(*Quercus petraea* (Matt.) Liebl) and chestnut (*Castanea sativa* Mill.). Hornbeam is an early leafing tree species,
chestnut is a late species while oak is intermediate. Hornbeam and oak are present in both forests, while chestnut is
present in Orsay only (Table 1). For each species, we focused on healthy and dominant trees, except for hornbeam (an
understory species). We collected budburst observations from 2000 to 2022, which yielded a dataset comprising 5
populations and 103 population-years. In each population, we observed between 28 and 309 individuals (mean 90)
(Table 1).



**Table 1. Description of the phenological and meteorological datasets.**

| Phenology Site | Coordinate | Meteorological station | Coordinate | Species | Number of year | Number of data | Number of trees (min / max / average) | Observation years |
|---|---|---|---|---|---|---|---|---|
| Orsay | 48.705° N, 2.165° E | Gometz-le-Châtel | 48.677° N, 2.136° E | *Quercus* | 23 | 153 | 29/190/85 | 2000-2022 |
| | | | | *Carpinus* | 20 | 124 | 29/146/50 | 2002-2006, 2008-2022 |
| | | | | *Castanea* | 21 | 112 | 29/192/80 | 2000-2007, 2010-2022 |
| Barbeau | 48.476° N, 2.780° E | Châtelet-en-Brie | 48491° N, 2.802° E | *Quercus* | 20 | 87 | 29/309/154 | 2003-2022 |
| | | | | *Carpinus* | 19 | 64 | 28/241/114 | 2004-2022 |

## 2.2 Phenology dataset

A team of eight local observers (including most of the authors of this paper) conducted the observations of developing buds in the tree crowns throughout spring. The observers used binoculars and occasionally received training in order to reduce observer bias (Liu et al., 2021). The interval between phenological observations was of 4 days on average (from 2 to 7 days). A tree was considered to have burst its buds when at least 50% of the buds in the upper third of the crown presented leaves that extended beyond the tip of the scales, which corresponded to stage BBCH 9 (Meier, 1997). At each observation date, we calculated the percentage of trees that had reached budburst in the tree population, dividing the number of trees at BBCH 9 by the total number of trees observed on that date and multiplying the result by 100.

## 2.3 Temperature data

We obtained the mean daily temperature data from the meteorological station nearest to the study sites (Table 1). However, there were missing values in the temperature data collected from the stations, especially before 1970. To fill these gaps and predict the missing data in order simulate budburst in previous years, we used the SAFRAN reanalysis data (grid-resolution of 8*8 km²) (Vidal et al., 2010), which we de-biased by establishing a linear regression between the local and corresponding SAFRAN temperature data from September of previous year to June.

## 2.4 Model description

We introduce a novel model, named the within-population variability (WPV) model, which was constructed to predict the progress of budburst in tree populations (i.e., percentage of trees having burst buds at a given date in a tree population). We hypothesized that the difference between individuals in the population was reflected in the difference of the forcing accumulation requirement ($F^*$).

We built the WPV model by modifying a state-of-the-art process-based model that simulated a discrete budburst event (i.e., budburst of an individual plant or mean budburst date in a tree population) (Lundell et al., 2020). In short, the model represents the release of endodormancy through the accumulation of "chilling" temperatures and simulates the ontogenetic growth of buds through the accumulation of "forcing" temperatures. One particularity of the model is that ontogenetic growth is regulated by the state of rest break and the prevailing temperature (Lundell et al., 2020; Hänninen, 1990; Hänninen and Kramer, 2007; Vegis, 1964). The ontogenetic competence, *Co* (a dimensionless [0, 1] multiplier),





is applied to represent this regulation (Lundell et al., 2020; Hänninen and Kramer, 2007; Hänninen, 2016). In the model,
budburst is considered to occur at date $t$ when a given sum of the forcing temperature is reached such that $F(t) \geq F^*$.
In the WPV model, we assumed that $F^*$ followed a normal distribution at the level of the tree population (see Fig. S1
for a flow chart of the model). At each date t, the model simulates the proportion of the population (BP, for *budburst*
*percent*) that has fulfilled the forcing accumulation requirement:
$$F^* = (\mu, \sigma^2) \qquad \text{eq.1}$$
$$BP(t) = 0.5 \times (1 + \text{erf}\left(\frac{F(t)-\mu}{(\sigma \times sqrt(2))}\right)) \times 100 \qquad \text{eq.2}$$

Where $F(t)$ is the forcing degree-day accumulation reached on day $t$, $\mu$ is the mean of normal distribution, $\sigma$ is the
standard deviation of normal distribution, and *erf* is the Gaussian error function.

The forcing accumulation $F(t)$ is calculated as the integral of a "forcing rate" as follows:
$$F(t) = \sum_{d=270}^{t} Rf_{act} \qquad \text{eq.3}$$
Where d is the start date of forcing accumulation ($d$ = DoY 270 in the previous year). In this model, the stage of
dormancy release and the stage of ontogenetic growth can occur simultaneously (i.e., the model belongs to the "parallel"
model category) (Hänninen, 2016; Chuine and Regniere, 2017). However, the forcing rate $Rf_{act}$, which is the actual
rate of ontogenetic growth, is affected by both temperatures and ontogenetic competence ($Co$). It is calculated as
follows:
$$Rf_{act}(t) = Rf(t) * Co(t) \qquad \text{eq. 4}$$
Where $Rf(t)$ is the potential rate of ontogenetic growth at time $t$, and $Co$ is the ontogenetic competence at time $t$; these
two variables are calculated as follows:
$$Rf(t) = \begin{cases} 0, & T(t) < T_b \\ T(t) - T_b, & T(t) \geq T_b \end{cases} \qquad \text{eq. 5}$$
Where $T_b$ is the temperature threshold (°C) above which forcing accumulation occurs.
The ontogenetic competence $Co$ varies over time and is simulated as:
$$Co(t) = \max\left(0; \min\left(1; g \times T(t) + h + \frac{Sr(t)}{100} * (1 - h)\right)\right) \qquad \text{eq.6}$$
Where $Co(t)$ is the ontogenetic competence at time t in the range [0, 1], which modulates the effect of the state of rest
break on the rate of ontogenetic growth (see Fig. S2). When $Co$=0, ontogenetic growth is stopped. The ability of
ontogenetic growth is restored between $Co$=0 and $Co$=1 with rest breaking. Finally, $g$ and $h$ are parameters (Lundell
et al., 2020), $T(t)$ is the daily mean temperature, and $Sr(t)$ is the state of rest break at time $t$, which is calculated as
follows:



$$Sr(t) = C_{tot}/C_{cri} \qquad \text{eq.7}$$
Where $C_{cri}$ is the chilling requirement for rest completion, and $C_{tot}$ is the actual accumulation of chilling temperature,
quantified as the number of chilling units (in chill units C.U.) and calculated from DoY=270 of the previous year up
to time t as follows:
$$C_{tot} = \sum_{d=270}^{t} Rc \qquad \text{eq.8}$$
Where the daily rate of chilling accumulation ($Rc$) is calculated as follows:
$$Rc = \begin{cases} 1, & T(t) < T_c \\ 0, & T(t) \geq T_c \end{cases} \qquad \text{eq.9}$$
Where $T_c$ is the temperature threshold (°C) below which chilling accumulation occurs.
**2.5 Parameter estimation**
We calibrated the model using budburst data obtained during the period 2000-2016 in Orsay (all three species:
hornbeam, oak, chestnut) and then validated it using data from 2017-2022 in Orsay (three species) and from 2000-
2022 in Barbeau (two species: hornbeam and oak). The model was therefore calibrated over 17 years for the three
species (Orsay populations, representing 52, 71 and 50 observation dates for hornbeam, oak and chestnut, respectively)
and validated over 29 site-years for hornbeam and oak (representing 89 and 114 observation dates, resp.), and 6 years
(29 observation dates) for chestnut. A previous study (Vitasse et al., 2009b) provided evidence of similar apparent
phenological responses to temperature among populations of the same species located as far as 650 km apart, which
also suggests the low differentiation of phenological traits across populations. Orsay and Barbeau populations are
separated by a distance of 50 km and experience a similar climate. This is why we used the Barbeau data as a validation
counterpart to the Orsay data used for calibration. The model predicts the percentage of budburst in the population
(from 0% to 100% budburst) along with the corresponding date. Thus, we calculated the root mean square error (RMSE)
over two dimensions (Fig. S3). First, we calculated RMSE over the percentage of budburst in the tree population (i.e.,
comparing the difference between the observed and predicted budburst percent occurring on the same day of the year,
DoY).
$$RMSE_{BP} = \sqrt{\frac{\sum_{i=1}^{n}(\sqrt{num} \times (BP_{obs,i} - BP_{pred,i})^2)}{\sum_{i=1}^{n}\sqrt{num}}} \qquad \text{eq.10}$$
Where $RMSE_{BP}$ is the root mean square error for budburst percent (expressed in percent), $num$ is the number of trees
observed on a given day of the year, $BP_{obs,i}$ is the observed percentage of budburst of datum $i$, $BP_{pred,i}$ is the predicted
percentage of budburst of same datum, and $n$ is the total number of data (e.g., n=50 in a hypothetic case where the
percentage of budburst has been observed five times per year on average over 10 years in a given population). We used
$\sqrt{num}$ as a weight in the calculation of squared errors to compensate for the fact that a very large number of trees
(i.e., >300 trees) were observed at some dates: these observations are more representative of the actual percentage of
budburst in the population (as compared to observations established for a smaller number of trees), although they also
tend to overrepresent them in the calculation of errors.



We then calculated the RMSE of dates (i.e., comparing the difference, in number of days, between the observations
and predictions for the same percentage of budburst; Fig. S3).
$$RMSE_{DoY} = \sqrt{\frac{\sum_{i=1}^{n}(\sqrt{num}\times(DoY_{obs,i}-DoY_{pred,i})^2)}{\sum_{i=1}^{n}\sqrt{num}}} \quad \text{eq.11}$$

Where $RMSE_{DoY}$ is the root mean square error for the budburst date (in days), $num$ is the number of trees observed,
$DoY_{obs,\,i}$ is the observed date of budburst of datum $i$ (e.g., the date when we observed 24% budburst for the population
of interest in a given year), $DoY_{pred,i}$ is the predicted date of budburst of the same datum (e.g., the date when the model
predicted 24% budburst in the same tree population and year), and $n$ is the total number of data.
Finally, we calculated the total RMSE as follows:
$$RMSE_{tot} = RMSE_{BP} + RMSE_{DoY} \quad \text{eq.12}$$

In the calibration stage, we determined the best parameter set as the one that minimized $RMSE_{tot}$.
In addition to RMSE, we also used mean bias error to evaluate the model forecast accuracy (in terms of budburst
percentage or DoY), which is calculated as follows:
$$mean\ bias = \frac{1}{N}\sum_{i=1}^{N}(obs_i - pred_i)$$

Where $obs_i$ and $pred_i$ are the i-th observation and prediction, respectively, N is the number of observations.
**2.6 Evaluating the modelled $F^*$ distributions**
To validate the modelled $F^*$ distribution, we simulated the distribution of the forcing accumulation at the date of each
BP observation. Because there are different observed BP in each year. We binned the observed BP data into 11 groups
(i.e., BP0, BP10, BP20…BP100, e.g., we regard the data between BP5 (date at which 5% trees burst buds) to BP15
(date at which 15% trees burst buds) as group "BP10"; note that BP0 refers to dates at which 5% or less trees have
burst buds, and BP100 refers to dates at which 95% or more trees have burst buds). Then we used a sigmoid function
to simulate the relation between BP and averaged corresponding forcing accumulation across all the years. We also
calculated their first derivatives (i.e., the increasing of BP per unit of forcing accumulation). Moreover, we calculated
the distribution of observed BP across all the years.
**2.7 Evaluating the response of the within-population variability of budburst to climate warming**
We used our model to predict budburst in the past (1961-2022) using historical daily mean temperature data and gap-
filled data using debiased SAFRAN reanalysis of temperatures (see above).
As explained earlier, our model simulates the percentage of budburst in a tree population at a given date. To evaluate
the response of the WPV of budburst to climate warming, we focused on the particular dates at which 20% and 80%
of trees in a given population had reached budburst (termed BP20 and BP80, respectively) and the duration between
these two dates (DurBB = BP80-BP20), which we consider to represent the variability of budburst within the





population for a given year. BP20 represents the "beginning" of budburst in the tree population, whereas BP80
represents its "end." We chose these quantiles instead of more extreme quantiles of distribution (e.g., 5% and 95%),
because they are well represented in our dataset (Fig. 1), thus implying higher model accuracy. For sake of model
evaluation, we calculated the DurBB in observed phenology data. Specifically, we selected years which had records
before BP20 and after BP80. Then the date of BP20 or BP80 were calculated by using the nearest two data (one is
below BP20 or BP80, another is above BP20 or BP80) through interpolation (e.g., 15 % budburst percent is on DoY
80 and 25 % budburst percent is on DoY 84. We can obtain the date of BP20 by interpolation, that is DoY 82).

### 233    2.8 Statistical analyses

For each population, we quantified by linear regression the sensitivity of budburst date (BP20 and BP80) and the
DurBB to time (days year$^{-1}$) and to Jan-May temperature (days $°C^{-1}$). Analysis of Variance (ANOVA) was used to
analysis whether the significance of the regression slopes ($\alpha$=0.05). All simulations and statistical analyses were
carried out with R statistical software v.4.0.3 (R Development Core Team, 2020).



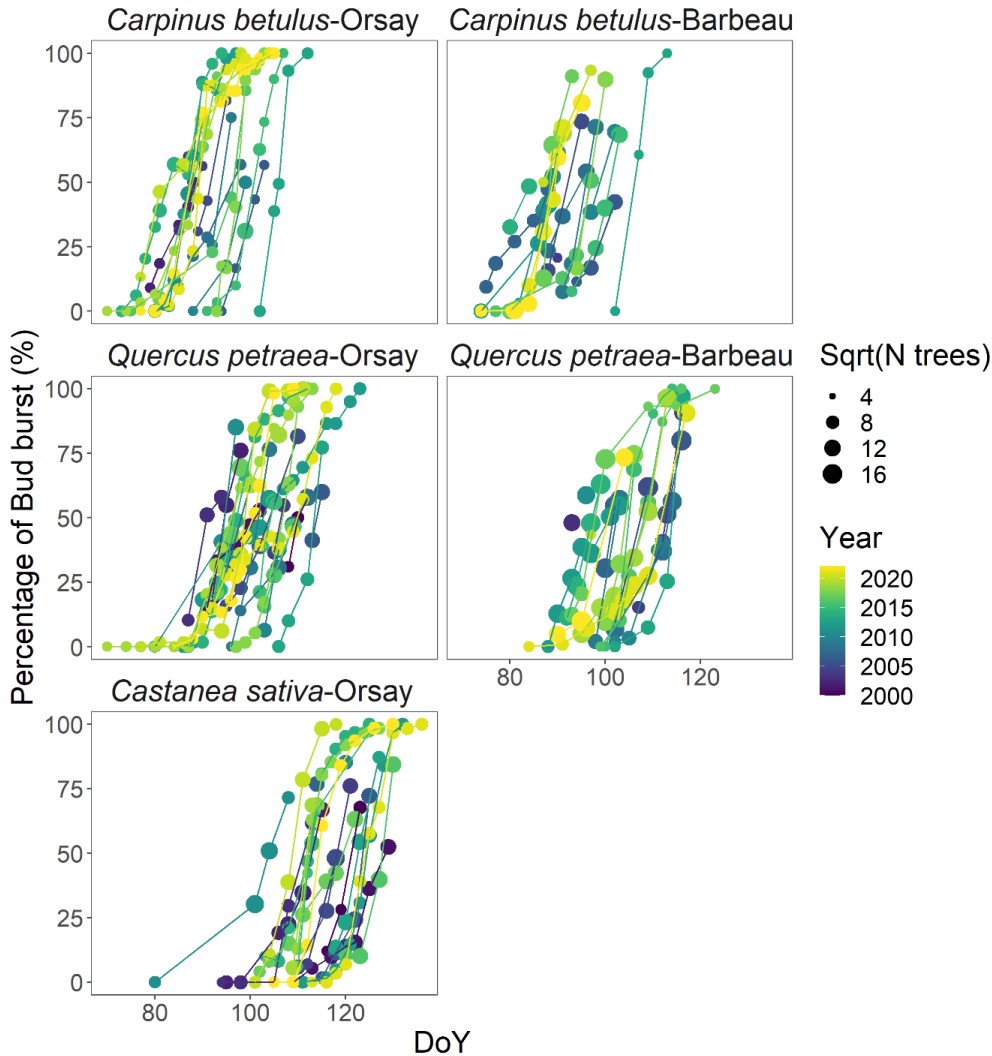


**Fig. 1. Observed percentage of budburst in five tree populations during the period 2000-2022. The size of the points is scaled**

**with the square root of the number of trees observed. The lines connect the dates of the same year.**
**3.  Results**
**3.1 Phenological observations**
Figure 1 shows the observed percentages of budburst in the five tree populations monitored from 2000 to 2022. These
percentage data were established based on 48,442 observations of budburst collected from individual trees. Among the
species, hornbeam was the earliest to reach budburst, typically over DoY 70-100, followed by oak over DoY 90-110,
and finally, chestnut over DoY 100-130. The budburst dates of the oak and hornbeam populations at Barbeau and





Orsay were very close, with average differences of 2 and 1 days (Table S1). The duration of budburst in the population
(DurBB) (i.e., time interval, in days, during which the proportion of trees having reached budburst increases from 20%
to 80%) differs for each species depending on the site and year, with a mean of 8 days over the whole dataset and
ranging from 3 days for hornbeam at Orsay in 2018 and 2021 to 21 days for oak at Orsay in 2012 (Fig.1).

## 3.2 Model performance

For all the populations considered here, the WPV model predicted with good accuracy the progress of budburst in tree
populations during spring as well as the interannual variability of budburst (Fig. 2, Fig. 3; see Fig. S4 for a comparison
of observed and simulated time series). The model predicted the percentage of budburst in tree populations with an
error ($RMSE_{BP}$) of 16% $\pm$ 0.2% for the calibration dataset and 20% $\pm$ 2.1% for the validation dataset. This corresponded
to prediction errors for the date of budburst ($RMSE_{DOY}$) of 3.9 $\pm$ 0.6 days for the calibration dataset and 5.6 $\pm$ 0.3 days
for the validation dataset. This compared well to the time resolution of the phenological observations (3-7 days). The
mean bias was within 1 day (Fig. 3).



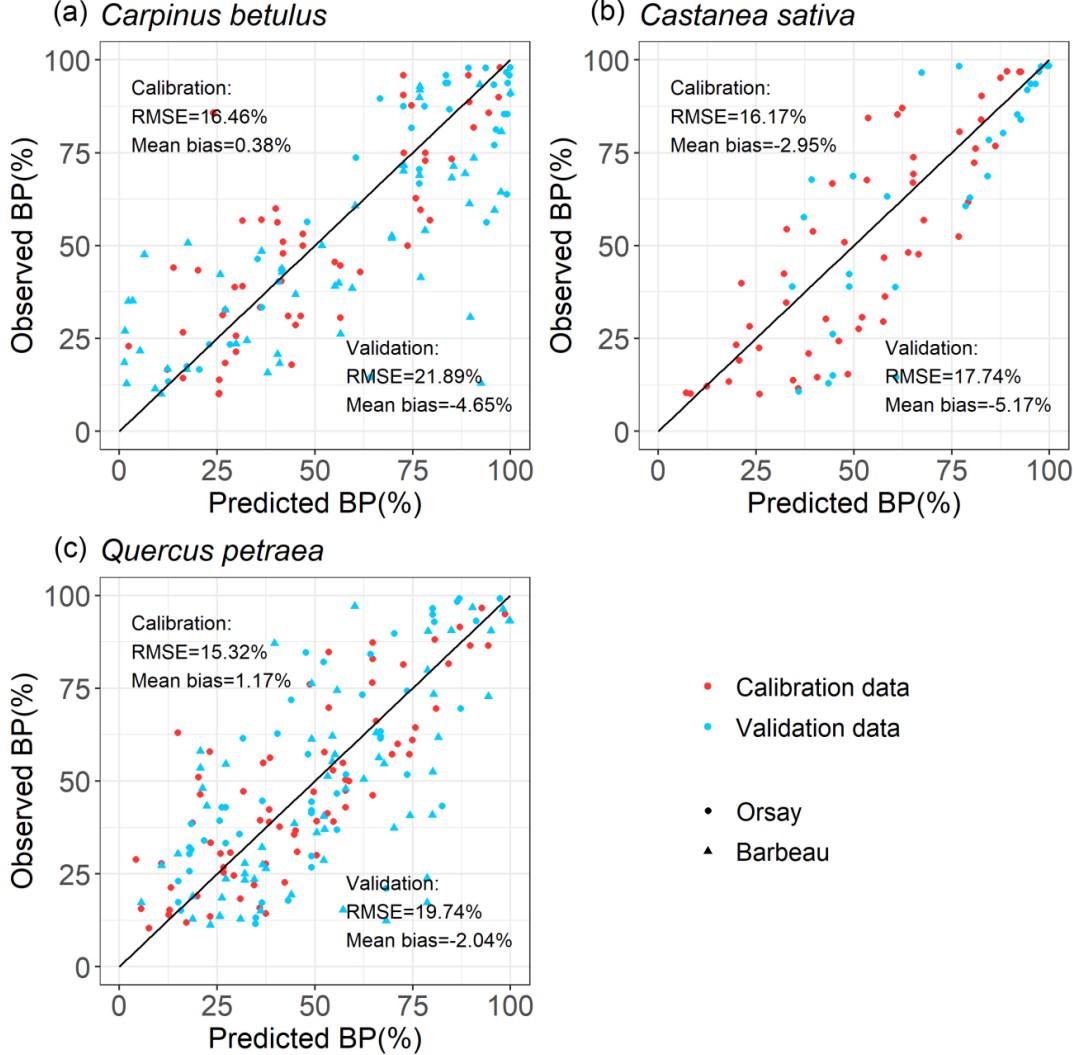


**Fig. 2. Evaluation of the within-population variability (WPV) model predicting the budburst percentage over calibration (red points) and validation (blue points) data. The points of circle are observed in Orsay and of triangle are observed in Barbeau. The points establish the correspondence between the observed and predicted percentage of budburst on an observation day in the population of interest. The one-to-one relation is shown as the black line. RMSE which is root mean square error for the budburst percentage and mean bias are shown. There are 52, 71 and 50 points (i.e., observation dates) for calibration and 89, 114, 29 points for validation for hornbeam, oak and chestnut, respectively.**



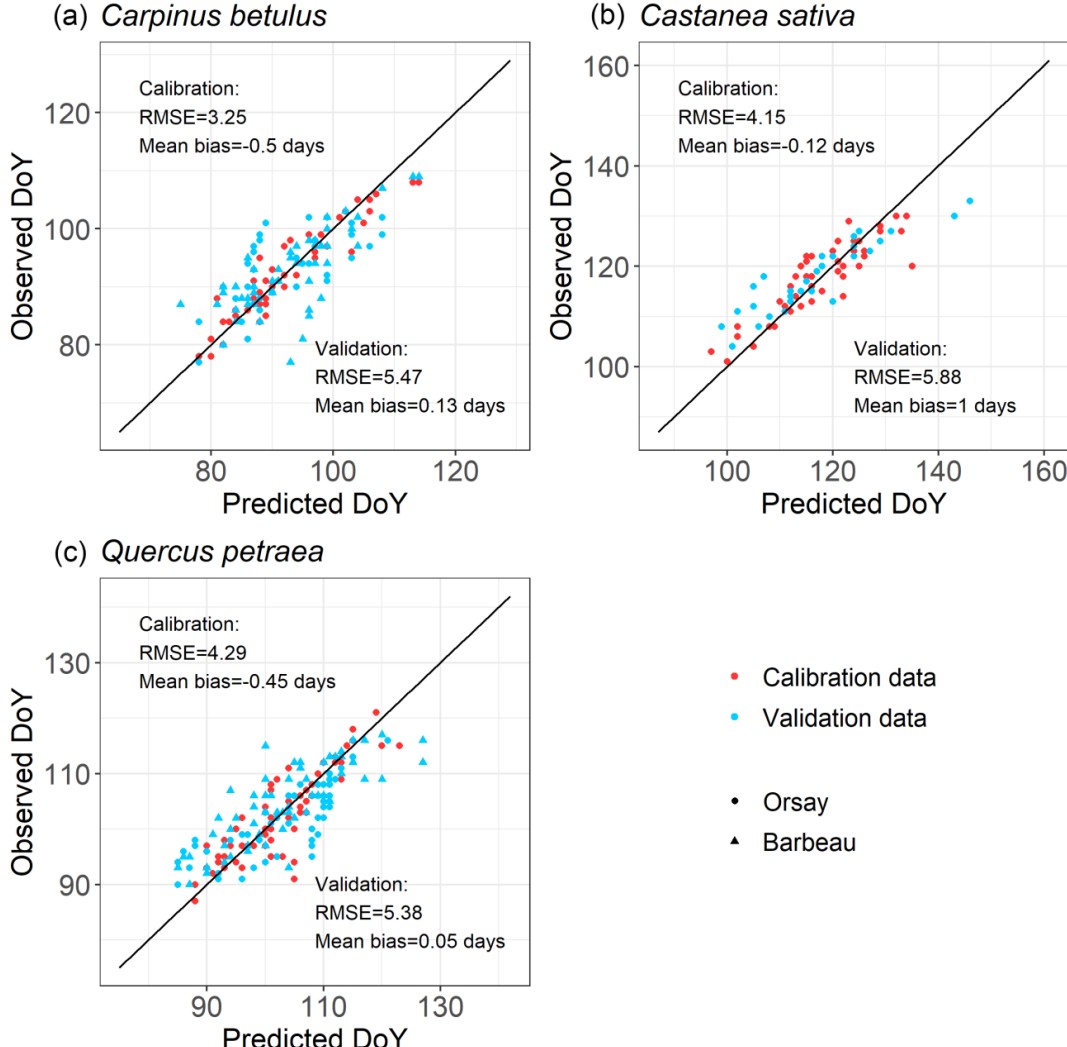

266

**Fig. 3. Evaluation of the within-population variability (WPV) model predicting budburst dates over calibration (red points)**
**and validation (blue points) data. The points of circle are observed in Orsay and of triangle are observed in Barbeau. The**
**points establish the correspondence between the observed and predicted budburst date on one observation day in the**
**population of interest. The one-to-one relation is shown as the black line. RMSE which is root mean square error for the**
**budburst percentage and mean bias are shown. There are 52, 71 and 50 points (i.e., observation dates) for calibration and**
**89, 114, 29 points for validation for hornbeam, oak and chestnut, respectively.**

### 3.3 Parameter variations across species

As mentioned earlier, we assumed that the forcing requirement ($F^*$) followed a normal distribution. The calibration
procedure yielded a set of distribution curves that differed across species (Fig. 4). We observed that the distribution of





$F*$ had a highest mean and standard deviation for oak compared with hornbeam and chestnut (Fig. 4, Table 2). The
distributions of $F*$ compared well to the actual distribution of forcing accumulation established from observations (Fig.
5b, e, h), validating the choice of the normal distribution. However, the modelled distribution did not overlap exactly
the distribution established from observed data, because the distribution of observations along the BP scale was uneven
(Fig. 5c, f, i). The temperature threshold for chilling accumulation ($T_c$) ranged from 9.7°C for chestnut to 10.5°C for
hornbeam and oak (Table 2). The temperature threshold for forcing accumulation ($T_f$) ranged from 3.9°C for hornbeam
to 7.7°C for chestnut (Table 2, Fig. S2). In all species, buds could not begin ontogenetic growth until the accumulation
of chilling to a certain extent (i.e., parameter h was negative for all populations, Table 2). We found that the threshold
of chilling accumulation necessary for the onset of forcing accumulation (i.e., value of Sr(t) from which Co becomes
positive) was very high for early species and decreased for late species (e.g., value of h increased approximately from
-0.98 in hornbeam to 0 in chestnut; see Table 2 and Fig. S2). Prevailing temperatures could compensate for the lack of
chilling accumulation (positive parameter g; Table 2) in hornbeam and oak, but not in chestnut (g=0).

**Table 2. Parameter values of the WPV model for three populations. μ (°C-days) and σ (°C-days) are the mean and standard**
**deviation of the distribution of F\*, respectively (Eqn. 1). $T_b$ and $T_c$ (°C) are the threshold temperatures for the accumulation**
**of forcing and chilling temperatures, respectively (Eqns. 5 and 9). g (°C$^{-1}$) and h (dimensionless) are the parameters**
**determining the interactive effect of the state of rest break and the prevailing air temperature on the ontogenetic competence**
**(Eqn. 6). $C_{cri}$ (number of days) is the chilling requirement of rest completion.**

| Species | Site | μ | σ | $T_b$ | $T_c$ | g | h | Ccri |
|---|---|---|---|---|---|---|---|---|
| *Carpinus* | Orsay | 138.4 | 29.6 | 3.9 | 10.5 | 0.0080 | -0.98 | 155.5 |
| *Quercus* | Orsay | 150.4 | 37.2 | 5.3 | 10.5 | 0.0032 | -0.89 | 153.0 |
| *Castanea* | Orsay | 131.2 | 30.7 | 7.7 | 9.7 | 0.0000 | -0.02 | 144.9 |







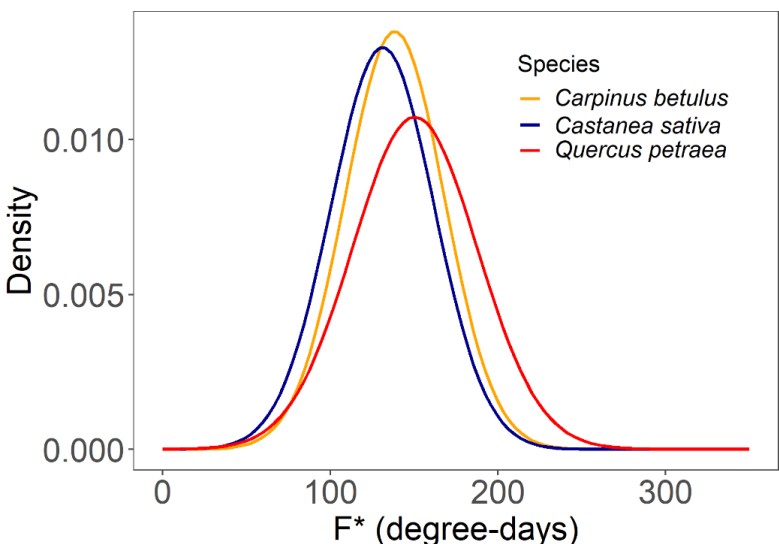


**Fig. 4. Normal distribution of the forcing requirement (F*) for three tree species.**

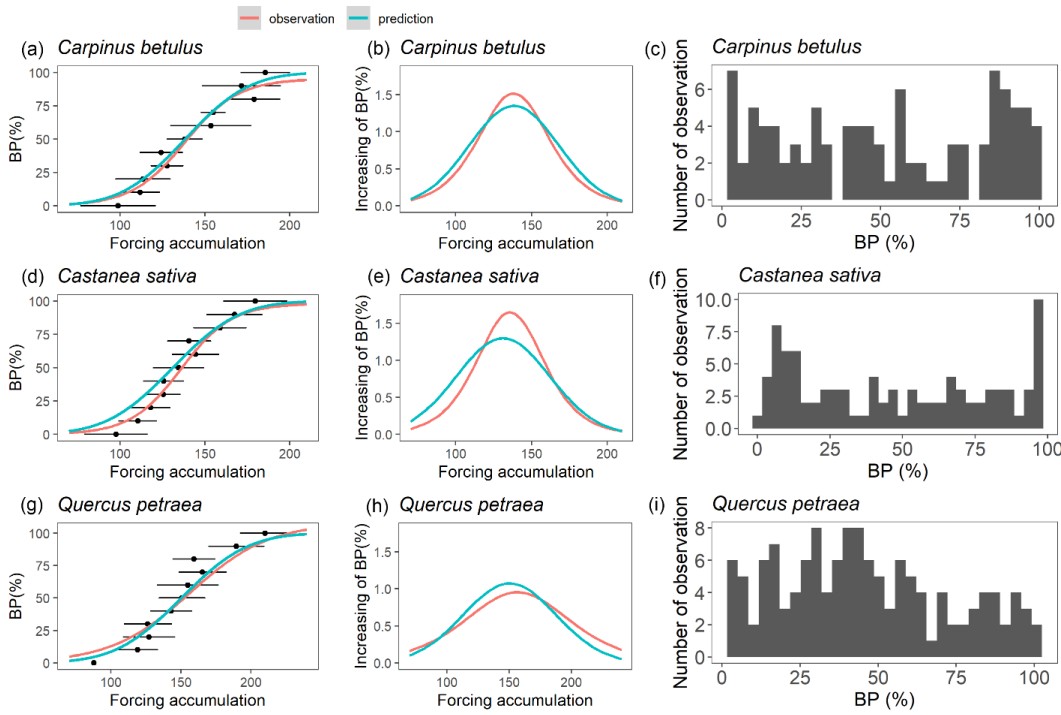


**Fig. 5. Evaluating the modelled F* distributions. Subplots (a, d and g) represent the relation between budburst percentage**
**(BP) and forcing accumulation. The black points and error bars represent the forcing accumulation required to reach a**





given budburst percentage in observed data (average across years ± one standard deviation). The red curves represent a
sigmoid function fitted to the black dots (a, d, g), and its first derivative (b, e, h). The blue curve represents predictions based
on the parameters in Table 2. Subplots (b, e and h) represent the increasing of BP per unit of forcing accumulation. Subplots
(c, f and i) show the distribution of observed data points in the budburst dataset.
**3.4 Retrospective analysis for within-population variability of budburst**
Over the past six decades (1961-2022), spring average temperature increased by +1.9ºC in Orsay and +1.4°C in
Barbeau (Fig. S5). Over this time period, our retrospective simulations suggest that the beginning (20%, BP20) and
end (80%, BP80) of budburst in tree populations has advanced significantly for all the species (Fig. 6), with
respectively 1.7 ± 0.6 days decade$^{-1}$ (mean ± SD across species) and 1.8 ± 0.4 days decade$^{-1}$ and apparent temperature
sensitivities of 5.8 ± 0.4 days ºC$^{-1}$ and 5.6 ± 0.4 days ºC$^{-1}$. These similar trends regarding the beginning and end of
budburst result in an unchanged duration of the budburst period (DurBB in the considered populations over the past
62 years (no trend in DurBB is significantly different from zero in Fig. 7, P>0.05). Notably, the interannual variability
of DurBB was large (Fig.6), and fairly simulated by our model (RMSE of 3.4 ± 1.8 days).

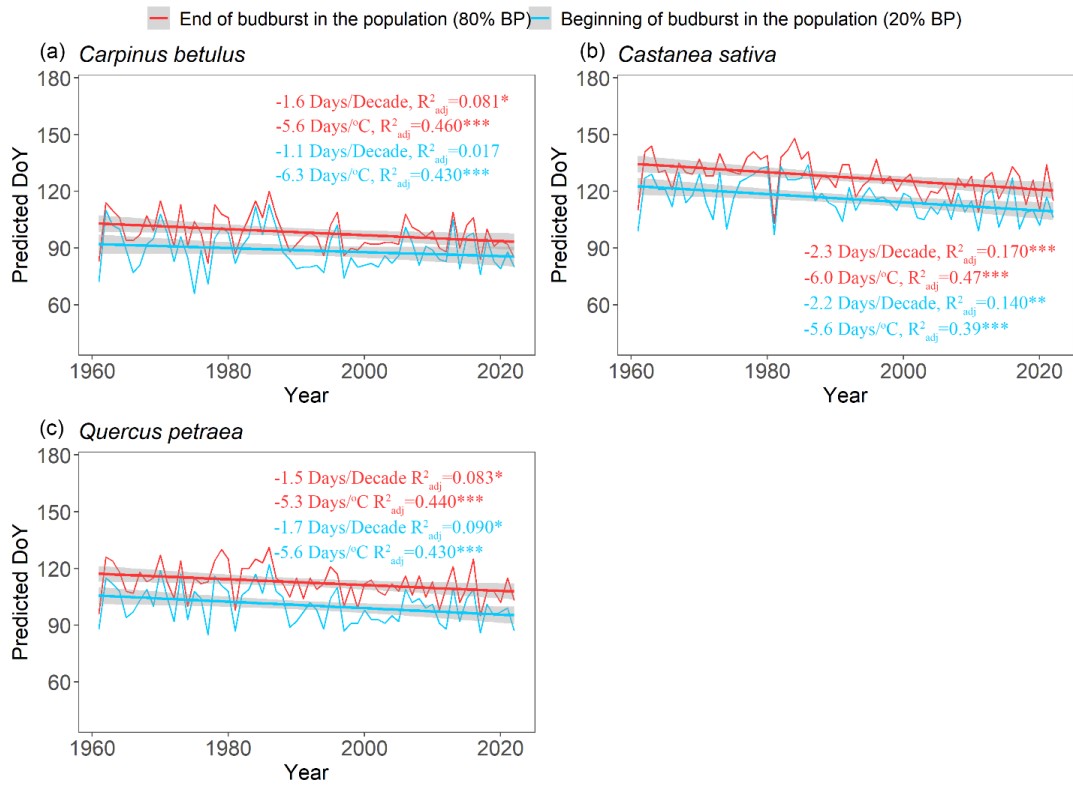


**Fig. 6. Simulated occurrence of the beginning (20%, BP20 in blue) and end (80%, BP80 in red) of budburst using the WPV**
**model for three tree species during the period 1961-2022. The fitted lines highlight the trends over the past 62 years. Text**
**in blue (red) shows the sensitivity of BP20 (BP80) to time and mean spring temperature (from January to May), respectively.**





**The sensitivity values are tested by linear regression analyses (\*: P<0.05, \*\*: P<0.01, \*\*\*: P<0.001) and adjusted coefficient**
**of determination (R²adj) is shown.**

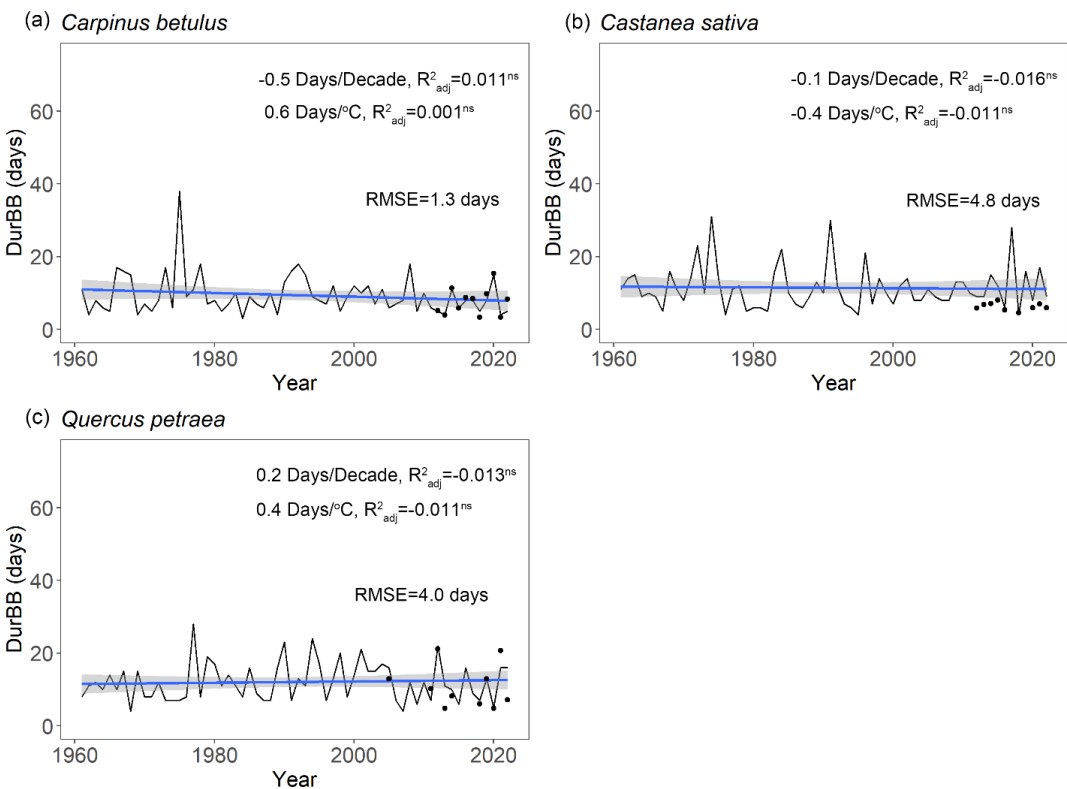


**Fig. 7. Simulated duration of budburst in the population (DurBB) using the WPV model for three tree species during the**
**period 1961-2022. The fitted line depicts the change in DurBB over the past 62 years. The sensitivity of DurBB to time and**
**mean spring temperature (from January to May) are tested by linear regression analyses (ns: P>0.05) and adjusted**
**coefficient of determination (R²adj) is shown. The black points are the actual durations of budburst observed in the data**
**(i.e., restricted to years when both BP20 and BP80 are available in a population).**
## 4. Discussion
To the best of our knowledge, this paper presents the first model simulating the within-population variability of
budburst in tree populations. An added value of this model is that it can simulate the duration of budburst in tree
populations. The central hypothesis of the model is that $F^*$, the amount of accumulated forcing temperature required
for trees to budburst, follows a normal distribution in tree populations. The ability of the model to simulate the
dynamics of budburst over the calibration and validation data, as well as the good agreement between the observed
and the simulated $F^*$ distributions (Fig. 5), lend support to this hypothesis for all the species and populations considered.
Our model yielded RMSE for the validation data (5.4 to 5.9 days), which are close to the temporal resolution of the



spring phenology observation (from 2-7 days) and similar to the typical prediction accuracy of models simulating
discrete (i.e., population average) budburst dates (e.g., Basler, 2016).
The variability in the timing of budburst among individuals in tree populations is considered to be mainly determined
by genetic diversity (Bontemps et al., 2016; Delpierre et al., 2017; Jarvinen et al., 2003; Rousi and Heinonen, 2007;
Rusanen et al., 2003) followed by the influence of the microenvironment (Delpierre et al., 2017; Rousi and Heinonen,
2007). The phenological ranking of individuals is largely conserved in tree populations (Delpierre et al., 2017), leading
to the identification of "early," "intermediate," and "late" trees (Malyshev et al., 2022). Further, the distribution of
budburst categories is not uniform in natural tree populations, with numerous "intermediate" individuals and
comparatively fewer "early" and "late" trees  (Malyshev et al., 2022; Chesnoiu et al., 2009; Zohner et al., 2018;
Caradonna et al., 2014), which lends further support to a unimodal distribution such as the normal law. Our model
reproduces this phenomenon, with categories of "early," "intermediate," and "late" trees corresponding to increasing
values of $F^*$. This core assumption of the model is supported by previous empirical studies, which observe that the
variability of $F^*$ could represent the variability of budburst among trees (Langvall et al., 2001; Rousi and Heinonen,
2007). Nevertheless, we could have chosen to assign the variance among individuals to one or several other parameters
of the model, related to the fact that genetic variations may affect any of the plant traits determining the modelled
parameters. For instance, Gauzere et al. (2019) found that the temperature yielding mid-forcing during ecodormancy
($T_{50}$) was more sensitive than $F^*$ in the UniChill model, which suggests that this parameter is another good candidate
for identifying the phenological behavior of individual trees in a population. Thus, we constructed a model assuming
that the threshold for forcing temperature ($T_b$, i.e., parameter of our model analogous to $T_{50}$) followed a normal
distribution, whereas $F^*$ was fitted as a constant parameter for the population. This model fitted the data less effectively
in both the calibration and validation steps (see Fig. S6 and S7 compared with Fig. 2 and 3), which further supports
our decision to assign the among-individual variance to $F^*$. Questions remain regarding the actual shape of the $F^*$
distribution. Indeed, natural selection can lead to traits that are not normally distributed (Caradonna et al., 2014), and
uneven distribution of observations may contribute to the non-perfect overlapping of observed and simulated F*
distributions (Fig. 5). However, earlier results (Vallet, 2020) showed that the form of the distribution had little influence
on the prediction accuracy.
We built the WPV model based on a two-phase parallel model framework, which describes the cumulative effect of
chilling and forcing temperatures on the endodormancy and ecodormancy phases, respectively (Hänninen, 2016;
Hänninen and Kramer, 2007; Lundell et al., 2020; Chuine and Regniere, 2017). This model structure is in line with
our current understanding of the physiological and molecular basis of dormancy in which the dynamics of the
dormancy mechanism are emphasized as opposed to a strict classification between the dormancy stages (Lundell et al.,
2020; Cooke et al., 2012). In this study, the threshold of chilling accumulation is up to 10.5°C for oak and hornbeam.
It is consistent with the experimental results in Baumgarten et al. (2021) which challenge the common assumption that
optimal chilling temperatures range ca. 4–6°C, showing 10°C is also effective for chilling accumulation in six dominant
temperate European tree species including oak. Furthermore, the model uses the concept of ontogenetic competence
($Co$) to simulate the process of regulation for the rate of ontogenetic growth by the state of rest break, a phenomenon
that has found support in phenological experiments (Lundell et al., 2020; Zhang et al., 2022). Our results demonstrate



that in the investigated species, *Co* is 0 until dormancy is released to a certain extent (Fig. S2), that is, ontogenetic
growth cannot start before a certain amount of chilling accumulation has been reached, which is consistent with
previous findings (Lundell et al., 2020; Zhang et al., 2022). According to the calibrated parameter values, ontogenetic
competence is also influenced by the prevailing temperature, although the effect is minimal. Indeed, parameter *g*,
which is related to the effect of the prevailing temperature, ranges from 0 to 0.0080 (Table 2), which is comparable to
values found in a previous study (Lundell et al., 2020). To some extent in this model, one consequence is that the effect
of the prevailing temperature can compensate for the deficiency of chilling accumulation.
Beyond introducing a model to describe the WPV of budburst in tree populations, our study aimed to quantify the
response of the duration of budburst (DurBB) to climate warming. We used temperature data to simulate the occurrence
of 20% (BP20) and 80% (BP80) budburst, and DurBB over the past decades. Our results suggest that the start and end
of budburst in tree populations have advanced over the past 62 years with climate warming (Fig. 6), which is consistent
with previous results showing advances in the population average dates of budburst (Wenden et al., 2020; Menzel et
al., 2006; Fu et al., 2015). In addition, our model simulates sensitivities of budburst to time and temperature that are
comparable to values reported earlier (Vitasse et al., 2009b, see Table S2). Our results point to significant sensitivities
to both time and temperature for oak as well as significant sensitivity to temperature for hornbeam, which is consistent
with the results of Vitasse et al. (2009b).
Our retrospective simulations suggest that there was not trend in the duration of budburst in tree populations, DurBB,
over the past 62 years (Fig. 7), in spite of climate warming (Fig. S5). Since both BP20 and BP80 advanced at a similar
rate, DurBB did not evolve over time over the 1961-2022 period. Interestingly, the analysis of temperature data
revealed no significant warming in the period of time from BP20 to BP80 over the past decades (P>0.05, Fig. S8). This
could explain why DurBB (time interval between BP20 and BP80) did not change over time, in spite of the strong
trends in both BP20 and BP80, caused by climate warming. Moreover, our study sites are located in the temperate
zone, at the heart (for oak and hornbeam) and at the north (chestnut) of our study species distribution areas (Caudullo
et al., 2017). At those sites, trees can accumulate enough chilling, or at least, chilling accumulation is not a limitation
for ontogenetic growth in nature so far, meaning that budburst is still advancing (Wenden et al., 2020; Piao et al., 2019).
Thus, the phenomenon by which DurBB increased with insufficient chilling accumulation in a given population (see
Zhang et al., 2021), their Fig. 2, 3, 4 for evidence in subtropical trees) did not appear in our retrospective simulations.
However, we can infer that if chilling accumulation can't be fulfilled under future, continuous climate warming, it will
take more time to fulfill the forcing requirement for late trees with a high forcing requirement, leading to the prolonging
of DurBB. A longer duration of budburst would increase the possibility of damage (i.e., freezing, insect damage).
These results suggest that the WPV of budburst should be given greater attention, because the longer duration of
budburst may be an important factor in the future when researchers project the damage in forests or determine the best
strategy for forest management.
We acknowledge that the projections of the WPV of budburst produced by the model are uncertain, first and foremost
because the parameter values were inferred from observation data collected in natural conditions as opposed to
controlled experiments (Hanninen et al., 2019). Another cause of uncertainty is the ability of the phenological response





of plants to acclimatize to the changing climate (Bennie et al., 2010). Under the hypothesis of plant acclimatization,
the parameters of the WPV model could have changed over the past decades, and would further change with ongoing
climate warming. Consequently, related experiments are urgently needed to improve our understanding of the WPV
of budburst to infer more reliable parameters and analyze the behavior of phenology models in different climates
(Hanninen et al., 2019). However, because our model addresses for the first time explicitly the within-population
variation of the physiological traits affecting phenology, it can contribute as a framework for future experimental
studies. In our study, we only considered the effect of temperature on budburst. However, other environmental factors
may also affect budburst (e.g., photoperiod). Previous studies showed that photoperiod is expected to modulate the
timing of budburst in late-successional species such as oak and chestnut, but not in early-successional species such as
hornbeam (Basler and Korner, 2012), but see a counter-example on oak in Malyshev et al. (2018). Moreover,
photoperiod may have a more complex interaction mechanism with temperature in terms of regulating the time of
budburst (Meng et al., 2021). We envision that improved versions of the WPV of budburst could be proposed based
on a more comprehensive understanding of the potential mechanism between phenology and environmental factors in
the future.

## 5. Conclusion

In conclusion, our work presents a novel model, simulating the continuity of budburst in tree populations in spring.
This phenological model can be adapted to the study of other stages of the tree phenological cycle, which are all of
continuous nature in tree populations (e.g., leaf senescence, wood formation etc.). We found budburst was advanced
in the past 62 years due to climate warming. However, the duration of budburst period of population was not affected
by increasing temperature. This is the first model simulating the within population variability of budburst in the
population. It provides a basis for implementation of a module in models directly interested in the within-population
variability of phenological and other functional traits (e.g., physio-demo-genetic models). It can also be used as a stand-
alone, to study the dynamics of phenological traits from the scale of individuals to the population and community in
the context of climate change.

## Code and data availability

The related phenology data and R code for the phenological model are openly accessible under
https://doi.org/10.5281/zenodo.7962840 and https://doi.org/10.5281/zenodo.7188160, respectively.

## Authors' contributions

ND and JL designed the research. ND, JL, AM, GV, DB collected phenological data. JL and ND performed the
research. JL wrote the manuscript with substantial inputs from all co-authors.

## Competing interests

The authors declare that they have no conflict of interest.



**Acknowledgements**
We acknowledge Eric Dufrêne for setting up the phenological surveys. We are also grateful to Eric Dufrêne and Jean-
Yves Pontailler for their invaluable contributions regarding the collection of phenological data. This work was
supported by the China Scholarship Council (202008330320).

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
