# Peer review of "A model of the within-population variability of budburst in"

_EGUsphere, 2023_

## Author Comment (AC1)

We thank both Dr Peaucelle and Pr Fu for their constructive comments, that helped us producing an improved version of the manuscript. We mentioned their contribution in the "acknowledgements" section of the manuscript (L. 471-472).

In the following response, our answers to reviewers appear in blue. The modifications made in the new version of the manuscript are highlighted in red, and duly referenced here with line number identification.

We hope the modifications we made to the manuscript will meet the reviewers' approval.

Summary:

In this study, Lin and co-authors developed a new phenology model to simulate the within-population variability of budburst in tree populations. Their model was calibrated and evaluated on 3 species and two French sites over 2000 -2022. Model simulations show good agreement with observations, both in simulating the average budburst date as well as the duration of budburst. The authors applied their model over 1961-2022 which simulated earlier budburst with climate warming, a result which is consistent with observations and current knowledge. Same simulations did not show significant evolution of the duration of budburst due to the lack of significant temperature increase during budburst.

Major comments:

The authors provide an interesting study and a new solution to address within population variability. Overall, the manuscript is clear and the methodology is appropriate. Results are well discussed in the light of the current knowledge. I only have some comments regarding the methodology and analyses that could strengthen the clarity and robustness of the analysis:

We thank Dr Peaucelle for his overall support for the manuscript, and for his detailed and constructive comments.

1)  The way the model was calibrated is unclear. At L. 205, the authors mentioned a calibration of model parameters by minimizing $RMSE_{tot}$. First, the optimization algorithm is not described here. I suggest the authors to describe that part (e.g. was it done with a gradient approach? Which package was used for that task? How a priori set of parameters were defined? How possible equifinality was handled?). Second, I believe there is a problem with the definition of RMSE from eq. 12 : $RMSE_{tot} = RMSE_{BP} + RMSE_{DOY}$. The authors sum two metrics with different units ($RMSE_{BP}$ is in %, $RMSE_{DOY}$ in days) which, as it is currently described, is wrong. Apart from the unit problem, minimizing a summed RMSE does not seem to be the best approach. If I understand, the goal here is to minimize both $RMSE_{BP}$ and $RMSE_{DOY}$. However, the same minimum $RMSE_{tot}$ can be achieved with multiple combinations of $RMSE_{BP}$ and

RMSE$_{DOY}$. I suggest the authors to have a multicriteria optimization instead of trying to minimize a combined metrics.

A1: We thank Dr Peaucelle for this suggestion. First, we have added the description about the optimization algorithm and the calibration procedure at L.208-211.

Our approach of using a multi-objective aggregate cost function is similar to approaches published earlier in the literature (e.g., Richardson et al. 2010, Keenan et al. 2011), now duly cited in the main text (L. 207). We do agree that this approach can suffer from the fact that the same minimum RMSE$_{tot}$ can be achieved with multiple combinations of RMSE$_{BP}$ and RMSE$_{DOY}$. For instance, the minimum RMSE$_{tot}$ can be achieved with low RMSE$_{BP}$ and high RMSE$_{DoY}$, or vice versa. We produced one supplementary figure and table to assess this. Figure S4 shows the relation between RMSE$_{tot}$ and RMSE$_{BP}$/RMSE$_{DOY}$. Table S2 shows the difference in prediction accuracy over one dimension (RMSE$_{BP}$ or RMSE$_{DOY}$) which is caused by using RMSE$_{tot}$. Using RMSE$_{tot}$ instead of RMSE$_{BP}$ or RMSE$_{DoY}$ yielded very similar results, with differences of less than 1% or 1 day when using RMSE$_{tot}$ instead of RMSE$_{BP}$ or RMSE$_{DoY}$, respectively.

2) To continue with the methodology, I wonder why the authors considered the Jan-May temperature to compute the sensitivity? It means that they account for extra temperature for early budburst compared to late budburst, which creates biases in the analysis. I would suggest the authors to considers the preseason temperature in order to avoid this bias (for example the average temperature of the month preceding budburst).

A2: We thank Dr Peaucelle for pointing out this question. We selected the Jan-May temperature because we aimed at comparing the temperature sensitivity of budburst in our dataset with the literature (we pinpoint here the work of Vitasse et al., 2009, who compared the sensitivity of budburst to temperature among populations in sessile oak using this period of time as a reference). Please see L. 409-412.

We do agree with Dr Peaucelle that considering pre-season temperature is a popular approach in the literature. Instead of the one-month period proposed by Dr Peaucelle, we calculated the budburst sensitivity to pre-season temperature considering a four-month period, that echoes the about four-month duration of the sequence from dormancy release to ontogenetic growth, as predicted by our WPV model. We added those results as a supplementary table (Table S4). The budburst sensitivity to temperatures are, as expected, always negative and significant (compare values in Table S4 and Fig. 6).

3) The authors hypothesized that F* follows a normal distribution (L. 274, L. 329) and try to validate that hypothesis with indirect comparisons to BP (%) in Fig. 5. This is a key aspect of their work and I believe that the distribution of F* can be directly

extracted from observations by computing F* for each observation. Having the "observed" distribution in F* would clearly strengthen the results and the discussion.

A3: We thank Fr Peaucelle for this suggestion. In fact, those "observed" distributions appear in the original manuscript, in Fig.5a, d, g. The black points and error bars represent the forcing accumulation required to reach a given budburst percentage in observed data (average across years ± one standard deviation), obtained by inference (i.e., calculating the F* "observed" when reaching a given value of budburst percent, with the other model parameters set at their calibrated value). Indeed, the true "observed" distributions of F* cannot be measured directly since F* depends on the value of the other model parameters. The red curves in Fig.5a, d, g represent a sigmoid function fitted to the black dots, and we also produced its first derivative in Fig.5b, e, h. As stated in the manuscript, "The distributions of $F*$ compared well to the actual distribution of forcing accumulation established from observations (Fig. 5b, e, h), validating the choice of the normal distribution. However, the modelled distribution did not overlap exactly the distribution established from observed data, because the distribution of observations along the BP scale was uneven (Fig. 5c, f, i)" (L. 295-298).

4) To continue with F*, the authors discuss L. 336-339, that the variability in the timing of budburst can result from the genetic diversity and microclimate. Following my previous suggestion regarding the computation of "observed" F* for each observation, looking at the evolution of F* for each individual could help in discussing that aspect. If the data allow it, showing if the same trees are always early/late within the population, or if it changes from year to year along with F* could help in quantifying that genetic vs. microclimate variability and provide some answers here.

A4: We thank Dr Peaucelle for this excellent suggestion. In fact, most of the observations from the Orsay populations were done on tagged individuals. We produced one figure showing the date of budburst and forcing accumulation in different years at tree scale in Orsay and discussed in L.363-365 (Fig. S7). This figure proves that tree individuals have a "phenological identity", i.e., maintain their phenological rank in the population, owing to underlying genetic, epigenetic and/or environmental differences among them.

5) It seems that the model is less performant for Castanea (Fig 5c and 7b) compared to other species. Is it linked to that null g parameter? DurBB seems to be well captured for the two other species and not Castanea, what could explain the variability in durBB then? Is it only linked to differences in F* distribution? or extreme temperature? Also the authors discussed the distribution in F*, could we imagine the same distribution for CCrit and what would it imply? Discussing these points could clarify the key message of the manuscript.

A5: We thank Dr Peaucelle for this comment. We do agree that the performance of model for Castanea is not as good as for the other two species when considering Fig.

5c and 7b. However, we would like to remember here that there are several possibilities. It should be noted our model only considered the effect of temperature. In fact, other environmental factors (e.g., photoperiod and soil water content) also can affect spring budburst. The model maybe improved including these factors. Because Castanea is latest species among these three species. Observation of budburst for Castanea is harder than for other two species. Because observation will be obstructed/affected by the crowns of other trees during observation for Castanea.

Contrary to Dr Peaucelle's hypothesis, the fact that the model is not as accurate for Castanea as for the other species does not stem from g being null. Null g parameter means there is no effect of prevailing temperature on ontogenetic competence (Co). We have tested the model performance with this effect of prevailing temperature on Co and without this effect for Castanea. The model without this effect is better.

We do agree with that the prediction for DurBB for Castanea is not as good as that for other two species. The original reason maybe the accuracy of observation for Castanea. Moreover, we did not have the exact records for the start (BP20) and end (BP80) of budburst in the population. Thus, the date of BP20 or BP80 were calculated using the nearest two data (one is below BP20 or BP80, another is above BP20 or BP80) through interpolation (e.g., 15 % budburst percent is on DoY 80 and 25 % budburst percent is on DoY 84. We can obtain the date of BP20 by interpolation, that is DoY 82). The interannual variability of DurBB is mainly caused by temperature, which is reproduced by our model. Our model is temperature-sensitive, which can simulate the DurBB based on different spring temperature scenarios. For example, if there is a sudden warm in early spring, the start of budburst in the population will be early, leading to larger DurBB. Or if there is a sudden cold in early spring, the situation will be the opposite. We have made a corresponding modification at L. 421-422.

We do agree that other parameters with same distribution are other choices. In fact, apart from F*, we have tried this method for the temperature threshold of forcing accumulation ($T_b$), the temperature threshold of chilling accumulation ($T_c$) and the requirement of chilling accumulation (Ccrit) (see L. 377-380). The performance of model which assume F* is normal distribution is best. And at present, there are some papers which have already shown the difference of budburst date between individuals by calculating forcing accumulation (see L. 82-85).

6) Finally, it would be very helpful to provide the scripts that were used to process the data, generate the analysis and figures for reproducibility of the results. The script currently archived on Zenodo provides the key functions to run the model but needs substantial effort to reproduce the results.

   A6: We thank for this suggestion. We have updated the code in order to provide more details. And we are pleased to answer any questions for the code or for the method of analysis.

I hope these comments will help the authors in improving their nice manuscript.

Best regards,

Marc Peaucelle

References cited:

Keenan, T. F., Carbone, M. S., Reichstein, M., & Richardson, A. D. (2011). The model–data fusion pitfall: assuming certainty in an uncertain world. *Oecologia*, *167*, 587-597.

Richardson, A. D., Williams, M., Hollinger, D. Y., Moore, D. J., Dail, D. B., Davidson, E. A., ... & Savage, K. (2010). Estimating parameters of a forest ecosystem C model with measurements of stocks and fluxes as joint constraints. *Oecologia*, *164*, 25-40.

**General comments**

The spring budburst in plants exhibits sensitivity to climate change. In this manuscript, the authors built the within-population variability (WPV) model based on a two-phase parallel model framework, to simulate the budburst in tree populations. Model performance of the WPV model has improved, rendering this research highly significant. I do like this study and it would be a nice contribution to the phenology modeling, and I have some concerns regarding the structure of the manuscript. Firstly, the introduction section need to be improved by adding the mechanisms of the dormancy release process and the introduction of the original model. Secondly, it is recommended to make appropriate deletions and adjustments to make the logic clearer for the whole manuscript, especially the content related to WPV in the introduction section. Additionally, the discussion section suggests adding species-specific effects on budburst and better explaining the reasons for considering intra-population variability to predict budburst. Therefore, a major revision is recommended.

We thank Dr Fu for his thorough review and detailed comments of our article. According to his suggestion, we modified the introduction (see L. 58-65, 70-76) and discussion (see L. 412-415, 428-431) in order to improve the logic of the text.

**Specific comments**

1. L28-30: Moisture (precipitation or air humidity) is also an important factor affecting the timing of leaf phenology in spring.

   A1: We do agree with this suggestion. We have added the sentence to show the effect of moisture on timing of leaf phenology in spring. Please see L30-33.

2. L55-67: These two paragraphs could be merged.

   A2: We thank for this suggestion and have modified this part. Please see L.58-65.

3. L62-67: 'the increasing warming rate during the budburst period'? Note that in the cited reference it is: 'slower warming rates during the budburst period'. Please carefully review and correct the citations throughout the manuscript to ensure they are accurately cited.

   A3: We thank for pointing this mistake. This part of content has been removed. And we have checked the references throughout the manuscript.

4. L137-140: Why is 'budburst percent' in italics? Also, 'erf' should be in italics in the formula.

A4: We thank for pointing this question. We have modified this mistake. Please see L.140-142.

5. L155-156: What does T(t) mean? Symbols that appear for the first time should be interpreted clearly.

A5: We thank for pointing this question. We have modified this mistake. Please see L.158.

6. L172-209: For the performance of the MPV model, please add R and p values to further verify the model accuracy. Also, what does 'i-th' mean in the L209.

A6: We thank for this suggestion. We have added the description of using R and p value to evaluate the model. "i-th" refers to the observation number *i* of a series of *N* observations. Please see L.220-222.

7. L251-258: Please add the values of R and p to the model performance and explain them.

A7: We thank for this suggestion. We have added R and p value to evaluate the model in the results. Please see L.268-273.

8. L378-386: Add some discussion about the effects of changes in BP20, BP80, and DurBB on the temporal niche, coexistence mechanism, and carbon sink capacity of tree populations.

A8: We thank for this suggestion. We have the discussion about the effect of changes in BP20, BP80 and DurBB. Please see L.412-415, 428-431.

9. L413-414: Other environmental factors affecting budburst should also include the interaction between environmental factors.

A9: We thank for this suggestion and have added the sentence to describe the effect of other environmental factors (e.g., moisture) on budburst. Please see L.442-448.

10. Table 1: Please correct '48491° N'.

A10: We thank for pointing this mistake. We have modified this mistake.

11. Fig. 1: 'Bud burst' in the title of the ordinate? Budburst.

A11: We thank for pointing this mistake. We have modified this mistake.

12. Fig. 2 and 3: Please add R and p values.

A12: We thank for this suggestion and have added R and p values in Fig. 2 and 3.

13. Fig. 6: What does each label represent? Please specify.

    A13: We thank for this question and have specified the meaning of labels in the figure title.

14. Fig. 7: The legend is missing. Also, what does each label represent? Please specify.

    A14: We thank for this question and have specified the meaning of labels in the figure title.

---

## Author Response (AR2)

We thank Dr Peaucelle for his constructive comment, that helped us producing an improved version of the manuscript.

In the following response, our answers to reviewer appear in blue. The modifications made in the new version of the manuscript are highlighted in red, and duly referenced here with line number identification.

We hope the modifications we made to the manuscript will meet the reviewer' approval.

The authors have clearly improved the clarity of the manuscript.

We thank Dr Peaucelle for his approval of our previous modifications.

I am still concerned about the aggregated RMSEtot metrics. I understand that this approach has been used in previous studies to which the authors refer. Nevertheless, I think that summing percentages with days is incorrect in the way it is presented in the manuscript.

For example, with RMSE doy = 10 days, if RMSE bp is represented between 0 and 1, RMSE tot will vary between 10 and 11. If the RMSE bp is represented between 0 and 100, the RMSE tot will vary between 10 and 110...

I would suggest to the authors to have the same units for RMSE doy and RMSE bp, as the two metrics will not have the same weight on RMSEtot.

A1: We thank Dr Peaucelle for this question. We agree with Dr Peaucelle that the unit of $RMSE_{DoY}$ and $RMSE_{BP}$ are different, which could be an issue. In order to solve this possible issue, we divided $RMSE_{DoY}$ or $RMSE_{BP}$ by the observation interval for budburst date or budburst percent, respectively. This is meant to scale the values of $RMSE_{BP}$ and $RMSE_{DoY}$, and attribute them comparable weights in the optimization procedure. We averaged the difference of DoY/BP between continuous observation for each species to obtain the observation interval based on our observation data. Then we calculated the new $RMSE_{tot}$ as followed:

$$RMSE_{tot} = \frac{RMSE_{BP}}{INT_{BP}} + \frac{RMSE_{DoY}}{INT_{DoY}} \qquad \text{(eq. 12 in the manuscript)}$$

Where $INT_{BP}$ and $INT_{DoY}$ are the observation interval for budburst percent and days, respectively.

$INT_{BP}$ and $INT_{DoY}$ measure the actual resolution of the observation data, and are thus the best achievable values in the optimization procedure. Hence, the new definition of $RMSE_{tot}$ represents the accuracy of the model in two aspects compared with observation interval. By definition of eq. 12, $RMSE_{tot}$ is now unitless.

The use of this new definition of $RMSE_{tot}$, as compared to the previous definition (where $RMSE_{tot} = RMSE_{DoY} + RMSE_{BP}$), had no influence on results for two species (Oak and Hornbeam: results were exactly the same). However, this new definition of $RMSE_{tot}$ yielded slightly different results for Chestnut. The new parameter set is actually closer to the ones obtained for the two other species. And the model predictions have generally improved for chestnut. The revised manuscript displays these new results (Fig. 2, 3, 4, 5, 6, 7; Table 2; Fig. S2, S4, S5, S7, S10; Table S1,S4, S5).

Apart from this point, I have no other comments to make on the manuscript.

Yours sincerely

Marc Peaucelle